# Cyberbullying as a Learned Behavior: Theoretical and Applied Implications

**DOI:** 10.3390/children10020325

**Published:** 2023-02-08

**Authors:** Christopher P. Barlett

**Affiliations:** Department of Psychological Sciences, Kansas State University, 492 Bluemont Hall, Manhattan, KS 66506, USA; cpb6666@ksu.edu

**Keywords:** cyberbullying, learning, theory

## Abstract

Cyberbullying perpetration has emerged as a world-wide societal issue. Interventions need to be continuously updated to help reduce cyberbullying perpetration. We believe that data derived from theory can best accomplish this objective. Here, we argue for the importance of learning theory to understand cyberbullying perpetration. The purpose of this manuscript is to firstly describe the various learning theories that are applicable to describe cyberbullying perpetration, such as social learning, operant conditioning, the general learning model, and others. Second, we delve into the Barlett Gentile Cyberbullying Model, which integrates learning postulates and distinguishes cyber from traditional bullying. Finally, we offer a learning perspective on interventions and future research.

## 1. Introduction

Technological innovation has paved the way for near instantaneous world-wide communication via the Internet. The adoption of the Internet, juxtaposed with (a) increased technology capabilities, (b) affordable software and hardware, and (c) accessibility, have changed nearly every sector in the industrialized world, such as education, medicine, banking, business, and others. As such, the Internet is used frequently by adults and youth alike. Recent survey data show that 97% of US youth [1] and 93% of US adults are online daily [2]. While most Internet behavior is likely unharmful, there are some who use the Internet for nefarious purposes, such as hacking, sending unwanted sexual depictions, and illegally downloading content. While continued empirical attention to these harmful online behaviors is needed, the purpose of the current paper will be on cyberbullying—defined as repeated unwanted and harmful behavior via online technology [3]. Results from a systematic review of the literature estimate the prevalence of cyberbullying perpetration to be between 6.3% and 32% [4]. 

Meta-analytic reviews have documented the correlations between myriad deleterious psychological variables (depression, anxiety, etc.) and cyberbullying perpetration [5]. It is imperative, therefore, that cyberbullying perpetration be reduced. One method to potentially decrease the frequency of cyberbullying perpetration is to better understand the variables and processes that predict its frequency to better inform intervention efforts. Results from several meta-analytic reviews suggest that cyberbullying perpetration-focused interventions are successful [6,7,8,9,10,11,12]. Although these meta-analyses differ in their scope, articles retrieved, publication date, effect size used, and other differences, the results all converge to suggest that cyberbullying perpetration interventions are effective.

The primary literature that was sampled and synthesized in cyberbullying intervention meta-analyses differ greatly in their theoretical perspective. For example, Media Heroes [13], iZ Hero [14], and a video intervention to reduce cyberbullying [15] all employ the theory of reasoned action/planned behavior in their lessons. Cyber-Friendly Schools [16] and Tabby [17] apply social ecological theory to cyberbullying prevention. Finally, ConRED [18] uses the theory of normalized social behavior to reduce cyberbullying perpetration. Overall, the application of existing theories to prevent cyberbullying is welcomed, and a recent burgeoning of empirical research that purports to predict cyberbullying perpetration and/or validate a cyberbullying intervention has adopted a wealth of theoretical perspectives. We contend here that many of the theoretical strides made in this literature and the research on intervention efficacy can be explained more holistically with learning theory. There are too few studies that have directly applied learning theory to understand cyberbullying perpetration, which negates using systematic literature review techniques to summarize the literature. Instead, our aim is to summarize the relevant literature that applies learning theory to cyberbullying perpetration. First, we will elucidate the learning theories relevant to cyberbullying perpetration. Second, we will apply learning theory to understand cyberbullying perpetration and extend those findings to interventions. Finally, we conclude with suggestions for future work. Our thesis is that cyberbullying is a learned behavior that becomes automated over time via experience and reinforcement. If our central claim is valid then, logically, it should follow that cyberbullying perpetration frequency can be reduced using similar learning postulates within an intervention context. 

## 2. Learning Theory

Before delving into the evidence that cyberbullying can be explained from a learning perspective, it is imperative to understand learning theory. That is, what processes and variables are important for social behavior to be learned? By learning, we mean, “…changes in the behavior of an organism that result from regularities in the environment of the organism” ([19]; p. 631). Historical reviews have thoroughly documented the evolution of learning research—from theory focusing on a very few specific processes to more general theories that integrate multiple processes [20]. Although we will not present an exhaustive list of all learning theories, we will focus on the theories that have application to cyberbullying.

## 3. Domain-Specific Learning Theories Related to Cyberbullying Perpetration

Domain-specific theories refer to a set of theoretical orientations that utilize a narrow set of concepts and processes while omitting others [21]. As outlined by Gentile and Gentile [22], there are myriad domain-specific theories that each emphasize learning differently, such as classical conditioning, discriminant learning, emotional learning, and others. We believe that all these theories have relevance to explain cyberbullying; however, data applying some of these domain-specific theories to cyberbullying are scarce due to the inability to measure cyberbullying (or cyber-aggression) at the state level. Indeed, most cyberbullying researchers use self-report questionnaires to assess how frequently cyberbullying behaviors have occurred across a given time-frame [23]—a type of trait measure that limits the application of certain learning theories in this domain. For example, emotional learning posits how state-based emotions can moderate the encoding and retrieval of information in memory, attentional processes, transfer of knowledge, and other cognitive processes [22]. Emotions—due to their reliance on affective episodes that foster current bodily changes, arousal, and appraisals [24]—are conceptualized as state (not trait) processes. Therefore, any application of emotional learning to cyberbullying is difficult due to mismatching measurement issues (cyberbullying as a trait and emotions as a state). We will focus our review on the learning theories that have empirical support. 

### 3.1. Operant Conditioning

As reviewed by Gentile and Gentile [22], the premise of operant conditioning is that behavior is shaped (learned) in response to reinforcement, punishment, and extinction processes. Reinforcement is focused on shaping behavior by either rewarding positive behaviors with some internal or external reward (e.g., giving a child money for good grades: positive reinforcement) or the removal of negative stimuli (e.g., allowing a child to not go to a boring movie for being nice to their siblings: negative reinforcement). In both cases, the goal is to increase a desired behavior. Punishment is focused on providing aversive consequences with the goal of stopping future negative behaviors, such as the removal of something positive after an unwanted behavior is enacted (e.g., giving a child a time out for hitting another child: negative punishment) or doing something negative as a consequence for a behavior (e.g., spanking a child for lying to their teacher: positive punishment). Finally, extinction occurs when the reinforcement or punishment no longer has an impact on shaping behavior (desired or unwanted). For example, taking away a toy after an unwanted behavior (negative punishment) is not likely effective for a 17-year-old teenager as it is for a 5-year-old child, but taking away a cellular phone is likely effective.

Empirical evidence suggests that cyberbullying perpetration can be explained via operant conditioning tenets. In a cross-sectional study of emerging adults, Barlett and Gentile [25] found that positive reinforcement by peers and family for harming others online positively correlated with self-reported cyberbullying perpetration—a finding also observed in a sample of Iranian adults [26]. In other words, if peers and family agree with and support cyberbullying actions, cyberbullying perpetration is likely. Conversely, research has shown that punishing youth for their cyberbullying others should curtail subsequent behavior. For instance, Legate et al. [27] sampled child–parent dyads and had youth complete measures of their cyberbullying behavior while the parents read a hypothetical vignette that described cyberbullying and responded to how they would address the cyberbullying in that vignette. Results showed that child cyberbullying behavior was negatively related to parent punishment in the hypothetical vignettes—as punishment increased, cyberbullying decreased. Moreover, research has shown that adolescent perceptions of informal sanctions (the likelihood that someone would be punished for cyberbullying others) negatively predicts cyberbullying perpetration [28]. 

Research focused on cyberbullying bystanders offers additional insight into the operant conditioning underpinnings of cyberbullying. Bystanders are those who are aware of cyberbullying and either do not intervene (termed a passive outsider), defend the victim (either online or offline), or reinforce the cyberbully (either online or offline) [29]. Longitudinal findings suggest that if a cyber-bystander is aggressive, then they are likely to engage in cyberbullying perpetration 8 months later [30], which is mediated by moral disengagement [31]. Several variables have been shown to predict one’s bystander role. Zhao et al. [32] showed that youth who have a history of cyberbullying others are more likely to be cyberbullying reinforcers or be uninvolved when witnessing other cyberbullying actions, whereas Van Cleemput et al. [33] showed that bystander youth who are empathic are likely to help and less likely to join the cyberbully. 

### 3.2. Social Learning

In the infamous Bobo Doll Study [34] researchers showed that children were more likely to punch, kick, and sit on an inflated Bobo doll after they viewed a model (real or filmed) acting aggressively. In short, individuals learn by witnessing how others behave. Additional empirical studies showed that social learning was observed when the model was rewarded for their aggressive actions (compared to punished [35]). Parents, peers, the media, and other entities model myriad positive and negative behaviors; a paucity of research has examined these influences on youth cyberbullying in a single study, but each individual source of learning has been tested. 

Focusing on peers, research has shown that social norms related to cyberbullying positively predict cyberbullying perpetration [36,37] and the willingness to join in cyberbullying others in a bystander role [38]. Indeed, meta-analytic findings confirm that social norms associated with cyberbullying correlate with cyberbullying perpetration [39]. However, much of this research assesses participant’s cyberbullying perpetration frequency and perception of social norms. One study showed a positive relationship between the participant’s and peer’s cyberbullying frequency [28]. These findings suggest that peer behavior and the perception of the acceptability of cyberbullying norms positively relate to cyberbullying perpetration. 

There has been an abundance of literature focused on the relationship between myriad parenting variables and cyberbullying. First, parenting styles have been shown to predict cyberbullying. Katz et al. [40] showed that cyberbullying was the highest for youth whose parents were perceived as controlling and inconsistent. Moreover, cyberbullying is higher when the parents are classified as neglectful or authoritarian [41]. Unfortunately, these results should be interpreted with caution due to the correlational of the data; however, longitudinal research confirms these findings—Wave 2 participant alienation mediated the relationship between Wave 1 parental flexibility and Wave 3 cyberbullying perpetration [42]. Second, research is mixed regarding the role that positive parental communication has on youth cyberbullying perpetration. Some studies have shown that open and positive communication between parents and youth negatively correlates with cyberbullying perpetration [43], whereas other studies have not shown this relationship [44]. Negative or problematic communication between parents and youth has been consistently shown to positively predict cyberbullying, which is mediated by psychological distress, social norms, and problematic social networking time [45]. Finally, research is mixed regarding the parental monitoring of cyberbullying behaviors. When youth perceive their parents monitoring their Internet time too much (e.g., having set time parameters for Internet time, parents checking websites, applying online filters, etc.) cyberbullying perpetration increases [46,47]. However, when that monitoring is less extreme—simply asking about the things youth do online instead of checking websites—then cyberbullying perpetration significantly decreases [48]. We believe that these findings can be partially explained by parental trust—higher levels of trust likely foster better and less restrictive online monitoring to decrease cyberbullying. Research has suggested that trust is correlated with cyberbullying [49]. 

Finally, research has shown that media exposure predicts cyberbullying perpetration; however, the type of effect (direct vs. indirect) is mixed. Supporting the direct effect of media violence on cyberbullying perpetration, longitudinal data suggest that media violence exposure predicts later cyberbullying perpetration [50,51], which seems intuitive since (a) cyberbullying is highly correlated with bullying and aggression [5] and (b) media violence exposure predicts aggression [52]. However, from a social learning perspective, it is unclear whether media violence exposure is directly related to cyberbullying perpetration. Indeed, we are unaware of any media program, movie, game, etc., that depicts cyberbullying perpetration in a positive light, which suggests that it is more likely that media violence exposure predicts more general aggressive tendencies that may manifest online [53]. In support of this indirect effect prediction, Zhu et al. [54] showed that normative aggressive beliefs mediated the relationship between media violence exposure and cyberbullying. More work is needed to fully disentangle whether the media violence to cyberbullying relationship is direct, indirect, or both. 

Overall, research from a social learning perspective has shown that cyberbullying is likely learned and reinforced from peers, parents, the media, and personal experiences. One caveat to this conclusion, however, is that there is a paucity of research investigating a direct social learning pathway from the source (parent, peer, media) to youth’s cyberbullying perpetration. For instance, a study in which parents and youth completed a measure of cyberbullying perpetration would allow for scholars to calculate the correlations between parent’s behavior (not the child’s perception of their behavior) and the child’s behavior. Future work is desperately needed in this area. 

### 3.3. Cognitive Learning

Theory posits that cognitive schemas—mental representations of people, places, and objects—and scripts—mental representations of events [55]—are developed via experiences (actual and vicarious), which are believed to reside in long-term memory. The creation, potential modification, and development of associations between myriad scripts and schemas is a result of cognitive learning. Script theory, for example, posits that once schemas are primed via myriad methods (reading words, viewing or listening to media, thinking of an object), they cause spreading activation that creates a network of related schemas to be active, and if a threshold of schemas in that network are active then numerous scripts will be activated and evaluated to guide subsequent behavior [56]. According to this theory [22], learning occurs when knowledge structures—scripts, schemas, cognitive biases, attitudes—are automatized and readily available for use. We contend that cognitive learning—because of the vast number of array of variables encompassed—has high applicability to cyberbullying research. This theory contends that cognitive learning via schema, scripts, and attitudes ultimately develops one’s personality [22], and an extensive amount of research in the cyberbullying domain has shown that myriad personality variables correlate with cyberbullying perpetration. Indeed, research has shown that cyberbullying perpetration is correlated with several learned aggressive knowledge structures, including normative aggressive beliefs, trait anger, moral disengagement, and others [5]. 

The integrative cognitive model of trait anger [57] posits that the interpretation of a hostile situation will ultimately predict anger and subsequent aggression due to (a) an increase in ruminative attention or (b) the inability to suppress anger via lack of effortful control. Li et al. [58] applied this theory to explain the cognitive basis for why childhood maltreatment predicts cyberbullying perpetration and showed that hostile attribution biases and anger rumination mediated these effects. Subsequent studies have shown the mediating role of other cognitive biases and variables that explain cyberbullying. For example, Li et al. [59] showed that belief in a just world mediated the relationship between parent–child attachment and cyberbullying perpetration, and Fan et al. [60] showed that self-esteem mediated the relationship between narcissism and cyberbullying perpetration. 

## 4. Integrative Learning Models Related to Cyberbullying Perpetration

Each individual domain-specific type of learning has its own merit and offers an interesting perspective on how social behaviors are learned. Furthermore, these, and other, learning theories are not in competition with each other, but, rather, may occur simultaneously in parallel with each other to compound the speed of acquisition and stability of the ensuing social behavior [22]. However, one criticism is that each offers a very narrow interpretation of how behavior is learned—each focusing on one specific process or mechanism while, at the very least, downplaying other mechanisms. Such limitations were one impetus for the development of a learning theory that integrated the myriad specific learning theories into a single comprehensive theory—the general learning model (GLM).

The general learning model is segmented into proximate (short-term) and distal (long-term) learning tenets [22]. The proximate GLM starts with personality (e.g., traits, genetics) and situational (e.g., the current environment) variables that either additively or interactively influence the sensation and then perception of a given stimulus. Sensation and subsequent perception are integral to the mental processing of elements in one’s social world, and can result in the discrimination of stimuli, classical conditioning processes, observational learning, and/or emotional encoding of stimuli. Therefore, a single initial exposure to a stimulus can instigate several specific learning processes that affect, and are affected by, one’s present internal state, which consists of inter-correlated affect and cognitions and physiological arousal. At this point in the model, multiple feedback loops exist to posit that the cognitive, arousal, and/or emotional experiences from stimuli reinforce the initial learning after stimulus exposure. Moreover, the internal state leads to higher-order appraisal and decision-making processes that eventually predict behavior. These processes start with an initial attribution about the social situation, and if one does not have the resources (time, motivation, cognitive energy) to devote to the environment, then impulsive behavior is likely, which may be highlighted via learned priming and heuristic processes. If one does have the resources to devote to an unsatisfactory appraisal, then re-appraisal processes are engaged to possibly change the initial attribution. Independent of whether re-appraisal is successful, thoughtful behavior is likely, which may be highlighted by cognitive learning processes.

The proximate general learning model incorporates myriad learning processes into a single theory, and research has shown that the GLM validly predicts myriad learned behaviors in the moment, such as helping behavior after prosocial media exposure [61]. Unfortunately, there is no research available to test the viability of proximate GLM processes extended to cyberbullying perpetration, due to measurement issues in cyberbullying discussed earlier. However, one can speculate that if an individual attacks another online for the first time, then the perpetrator will attend to and perceive selected elements regarding the cyber-attack (e.g., the content of what was typed, how many likes one received, etc.). This will likely cause changes to the internal state specific to cyberbullying, which may include: positive emotional outcomes from harming another and not being identified, positive reinforcement from others who share your opinions and sanction the cyber-attack germane to operant conditioning, the physiological arousal response of doing something harmful that maps onto classical conditioning, and others. Following this, GLM explicates that subsequent cyber-aggressive behaviors are likely in the short-term.

The general learning model also posits distal, or long-term, learning processes that predict the development of personality. Here, the GLM argues that continued exposure to social behaviors and the subsequent short-term learning postulates previously described will lead to changes to one’s personality through the development of several knowledge structures consistent with behavior. Specifically, continued exposure and practice with stimuli predict the development of cognitive-emotional constructs (i.e., attitudes and stereotypes), emotional constructs (i.e., affective traits, affective habituation), and cognitive constructs (i.e., beliefs and scripts), which yield the development of one’s personality. In other words, continued cyberbullying experiences are one possible impetus for the development and automatization of myriad cyberbullying and aggression-related scripts, schemas, biases, attitudes, etc., and the accumulation of these, and other, knowledge structures defines cyberbullying personality. 

## 5. Cyberbullying Perpetration as a Learned Social Behavior

Again, our central thesis is that cyberbullying perpetration is a learned social behavior. However, before delving into the evidence to support this thesis, an important question must be answered: What is the importance of understanding cyberbullying through a learning lens? We have noted elsewhere [62] that the best way to prevent antisocial behavior is to understand the psychological processes and variables germane to that behavior so that intervention efforts can be armed with the best possible information in order to be most successful. While multiple theoretical perspectives offer insights into cyberbullying that can accomplish the goal of understanding cyberbullying perpetration, a learning perspective offers important theoretical insights. For example, if social learning and operant conditioning postulates can be extended to cyberbullying, then interventions that focus on stopping positive reinforcement of cyberbullying actions should undermine the learning of cyberbullying attitudes, beliefs, and cognitions that likely predict its frequency. Moreover, a learning perspective highlights the need for interventions focused on parental, peer, and school entities in addition to the child’s education to prevent future cyberbullying perpetration. These and other examples highlight how learning theory as applied to cyberbullying perpetration can offer a unique application to prevention.

Can a learning theory, such as GLM, explain cyberbullying perpetration? We believe that it can. Figure 1 displays how we believe that cyberbullying can be explained by the distal GLM [63]. Three important points are noteworthy regarding Figure 1. First, GLM postulates can explain the development of cyberbullying through several learning processes and constructs. In Figure 1 we replaced the theoretical constructs of GLM depicted on the left with the labels of variables found in the literature on cyberbullying perpetration on the right (we denoted these additions in yellow for ease of readability). Second, there are still many questions remaining that have received little to no empirical attention. For instance, we are unaware of any research investigating the “chunking/encoding” learning that GLM classifies as a cognitive-behavioral construct, which we will elaborate on later. Third, it is likely that some of the learning processes elucidated in the GLM do not transfer onto cyberbullying well. For instance, GLM notes physical skill as a learning process, which dictates how continued practice with a behavior automatizes a behavior. Physical skill as a cognitive-behavioral construct is clearly applicable to myriad trained behaviors, such as driving a car or shooting a free throw in basketball; however, it is less clear how this transitions to online bullying. 

Evidence from empirical studies support the application of GLM to understand cyberbullying perpetration. We will elaborate on such transitions below:

### 5.1. Perceptual and Cognitive Constructs

There are three constructs within this section of GLM. The first is perceptual and expectation schemata, which describe cognitive knowledge structures regarding an individual’s behavior or behavior of others, and research has shown that cyberbullying is correlates with—and predicts—several of these knowledge structures. Perceptions of anonymity represent an individualized perceptual schema, in which individuals believe themselves to be less likely to be identified in online environments. Correlational [64] and longitudinal [65] findings show that cyberbullying is related to these perceptions. In an interesting study, Sticca and Perren [66] had youth read several hypothetical scenarios that described a peer who was excluded from a party, and the scenarios differed on whether the communication was delivered online or in person and if the message was anonymous or not. Participants then rated the scenario on aggression severity and humiliation, and results showed that the more humiliating and threatening scenario was when the rejection was online and anonymous. These findings can be best explained by online disinhibition theory [67], which posits that individuals are likely disinhibited in online (versus face-to-face) environments, which change the likelihood of aggressive behaviors (amongst other outcomes). For instance, the perceived anonymity afforded an online aggressor, juxtaposed with asynchronicity (the lack of real time interactions online), the minimization of status (absence of cues indicative of status or authority), and other constructs, likely increase cyberbullying [68]. Moreover, research has shown that cyberbullying perpetration correlates positively [69,70,71] with normative aggressive beliefs (NOBAG; the cognitive belief that aggression is acceptable after a perceived provocation [72]), and with cognitive interpretation of ambiguous situations as hostile [73], termed hostile attribution biases (HAB [74]). Moreover, GLM posits that several beliefs and scripts are important perceptual and cognitive constructs. One belief that has been shown to predict cyberbullying is the belief in the irrelevance of muscularity for online bullying (BIMOB). BIMOB is a belief theorized to be the consequence of cyber-aggression which emphasizes the common belief that anybody—no matter how physically small or weak—can harm others due to the online nature of cyberbullying [25]. Finally, research has shown that cyberbullying expectations (a behavioral script for the likelihood of future cyberbullying) is predicted by myriad variables that share variance with cyberbullying perpetration, such as moral disengagement, positive cyberbullying norms, and self-efficacy [75]. 

### 5.2. Cognitive-Behavioral Constructs

Gentile and Gentile [22] explicated several types of learned outcomes that are classified as cognitive-behavioral that form a function of continued practice. Evidence for learned cognitive-behavioral constructs can be seen by comparing experts to novices on some tasks. The translation of these GLM tenets to cyberbullying is less clear. While it is true that differences in conduct problems, prosocial behavior, and hyperactivity-inattention emerge between cyberbullies, cyber-victims, cyberbully-victims, and uninvolved youth [76], the conceptualization of an “expert cyberbully” is difficult. Moreover, the exact encoding and mental processing of cyberbullying-related information in the moment remains unclear. Therefore, we contend that the cognitive-behavioral portion of the distal GLM is in need of empirical attention and study. 

### 5.3. Cognitive-Emotional Constructs

Attitudes and stereotypes are the two constructs that are labeled as cognitive-emotional according to GLM [22]. There is a rich social psychological literature showing that attitudes predict behavior [77], and research has shown that positive cyberbullying attitudes—evaluating cyberbullying as positive and/or justified—positively correlates with cyberbullying behavior [78]. For instance, using a correlational design with US adults, Doane et al. [79] showed that cyberbullying attitudes significantly predicted three types of cyberbullying (deception, malice, and public humiliation) indirectly through cyberbullying intentions. Moreover, several correlational studies show that cyberbullying attitudes directly predict cyberbullying perpetration [80], whereas other research findings suggest an indirect effect of cyberbullying attitudes to cyberbullying perpetration through behavioral intentions [81]. Finally, there is a paucity of research examining the relationship between stereotypes and cyberbullying; however, scholars have theorized about such a link. Indeed, Keum and Miller [82] described a model of online racism, which posits that online anonymity perceptions predict online disinhibition (described earlier) to predict in-group biases and stereotype formation. We are unaware of any primary research validating this model, and future work is needed. 

### 5.4. Emotional Constructs

The final organizational section of the distal GLM consists of emotional constructs, which consist of several processes germane to changing personality as a function of repeated learning [22]. Affective habituation refers to the learned association between the behavior and emotional constructs (e.g., cyberbullying others is associated with excitement). Research employing the uses and gratifications framework [83] has shown that one possible motive for cyberbullying others is the entertainment and revenge that participants classified as cyberbully-victims (individuals who are both victimized and perpetrate online bullying) experience in harming others online [84]. Next, GLM includes conditioned emotions, which includes desensitization—the decreased emotional, physiological, and cognitive response to a stimulus [85], which can be operationalized via empathy [86]. Results from meta-analyses show that empathy is related to cyberbullying perpetration [5]. Finally, Gentile and Gentile [22] noted that affective traits—personality dimensions related to emotional expression—are conceptualized as an emotional construct within GLM. Research has shown that cyberbullying perpetration is correlated with several affective traits, including trait anger [87], neuroticism [88], and emotional intelligence [89]. 

## 6. Uniquely Predicting Cyberbullying

Despite the growing research support for GLM applications to cyberbullying, one caveat is that most learning theories are not specific to the online world—GLM included. Instead, both domain-specific and integrated learning theories can predict myriad antisocial behaviors, including cyberbullying, traditional bullying, and aggression. Indeed, one strength of GLM is its ability to describe and predict a plethora of behaviors, not just cyberbullying. Theoretical specification to online behavior is greatly needed in the cyberbullying domain because: (a) cyberbullying is specific to online mediums, making it theoretically distinct from traditional forms of bullying and aggression; and (b) cyberbullying interventions can be better informed with theory devoted specifically to online harm. Cyberbullying is unique because of the increased anonymity afforded to the online aggressor, the irrelevance of one’s physical stature, the non-physical nature of cyberbullying, the ability to have others see the online harm across the world at instantaneous speed, and other factors [90].

The theoretical gap in uniquely predicting cyberbullying incrementally from more traditional forms of bullying using learning-based underpinnings was filled with the validation of the Barlett Gentile Cyberbullying Model (BGCM [25]). Derived from the theoretical postulates of the general learning model, the BGCM posits that after youth cyber-attack another for the first time, they likely learn to perceive themselves as more anonymous online and learn that their physical stature is moot due to the online environment necessary for cyberbullying to occur (BIMOB). Continued cyber-attacks further develop and eventually automatize these perceptions and beliefs to form positive cyberbullying attitudes, which eventually become automatized to yield subsequent cyberbullying behavior. The importance of automatization and development highlight the learning emphasis of the BGCM. Figure 2 depicts the BGCM and highlights the learning stages that link initial cyber-aggressive actions to eventual development of cyberbullying propensities. 

The BGCM has received much empirical support. Indeed, research has shown that Wave 2 cyberbullying attitudes mediate the longitudinal relationships between Wave 1 anonymity and BIMOB with Wave 3 cyberbullying perpetration [91]. Importantly, BGCM postulates remain significant while controlling for traditional bullying perpetration [92]. This is important to show the incremental validity of the evidence that the BGCM predicts cyberbullying above and beyond traditional bullying, despite the high correlation between both forms of bullying perpetration [5]. Finally, research has shown that BGCM relationships are observed in countries across the world [93]. 

The core learning postulate of the BGCM is that each cyber-attack perpetrated is a learning trial, and longitudinal research has shown support for the learning tenets of BGCM—and by extension the GLM. For example, Barlett et al. [94] used a three-wave longitudinal design sampling Singaporean youth and found (a) strong stability over time for cyberbullying attitudes and cyberbullying perpetration, (b) early cyberbullying attitudes predicted later cyberbullying perpetration (consistent with BGCM), and (c) early cyberbullying perpetration predicted later cyberbullying attitudes (consistent with learning). In other words, cyberbullying attitudes reinforced cyberbullying perpetration, and, in turn, cyberbullying perpetration further reinforced cyberbullying attitudes. Moreover, Barlett and Kowalewski [95] used a four-wave longitudinal design with an emerging adult sample and showed that (a) Wave 1 anonymity perceptions and BIMOB predicted Wave 2 cyberbullying attitudes to predict Wave 3 cyberbullying behavior, consistent with BGCM, and (b) Wave 3 cyberbullying perpetration predicted Wave 4 anonymity perceptions and BIMOB, consistent with the learning theory. This is an important finding because this shows that continued cyberbullying perpetration—which was derived from BGCM tenets—further reinforces cyberbullying-related knowledge structures, supporting the feedback loop germane to BGCM processes.

## 7. Moderators in the Learning of Cyberbullying

The wealth of support for the BGCM validates the theoretical postulates and applied implications of this theory to understand and ultimately reduce cyberbullying. However, we do not believe that the current version of the BGCM depicted in Figure 1 is ubiquitous. Individual differences likely influence each observed variable in the BGCM framework and the underlying learning underpinnings of BGCM. Each will be briefly discussed.

First, research has shown substantial individual differences in cyberbullying perpetration. We have already elucidated several of these variables (in the cognitive learning section), but several other individual difference variables predict cyberbullying. For instance, meta-analytic results show sex differences in cyberbullying are moderated by age—males are more likely to cyberbully than females at older ages (emerging adult and above), but females are more likely to cyberbully than males at young ages (approximately 9–11 years old) [96]. Moreover, Barlett et al. [91] showed that emerging adult males had higher levels of anonymity perceptions and cyberbullying attitudes compared to emerging adult females. In addition to age and participant sex, other personality variables have been shown to moderate key relationships. Barlett et al. [97] used a cross-sectional study of US adults and showed that cyberbullying attitudes mediated the relationship between anonymity perceptions and cyberbullying perpetration (supporting BGCM); however, the relationship between anonymity and cyberbullying attitudes was moderated by dispositional fear of retaliation—a personality variable that measures the extent to which one is afraid of another’s retaliation and changes their behavior. Closer examination of this moderated effect suggests that people who perceive themselves as anonymous online are likely to endorse cyberbullying attitudes when they are fearful of another’s retaliation. 

Second, technological moderators likely influence BGCM processes. Technological moderators are those individual differences that are specific to technology, such as time spent online and technology access. Indeed, research using a cross-sectional design showed that time spent online predicted BIMOB and anonymity perceptions, and that perceptions of technology access correlated with cyberbullying attitudes [98]. Moreover, self-efficacy related to the ability to cyberbully others (e.g., having the ability to create and send a computer virus) correlated with cyberbullying perpetration [99].

We want to explicate that there is no published research testing the moderating influence of any of these variables in BGCM processes. Future work is needed to uncover what variables moderate the learning explicated by BGCM; however, the general learning model theorizing posits that biological (e.g., genes, sex) and environmental modifiers (e.g., internet access, SES, living conditions) influence the learning of social behaviors [22]. This theorizing gives precedent to study myriad moderating variables within the context of BGCM learning—despite the paucity of work in this domain. Figure 3 displays a conceptual model of the role that various moderators may have on BGCM. 

The moderating influence of myriad variables in the cyberbullying process has implications for intervention efforts. Within the context of cognitive learning, research has shown that school staff’s self-efficacy beliefs in intervening in a cyberbullying incident are moderated by school status—when self-efficacy beliefs are low, low-status staff are less likely to intervene compared to high-status staff [100]. In theory, if staff (or anyone) intervenes in a cyberbullying incident, then the perpetrator will likely not be positively reinforced for their online behavior to mitigate BGCM learning. 

## 8. Intervention Applications

Results from several meta-analyses showed that interventions derived to reduce cyberbullying perpetration are successful [8]. These interventions vary widely in their focus, populations that the intervention is normed around, the degree of cyberbullying reduction, the study design, and others. Cioppa et al. [101] reviewed several cyberbullying interventions and rated each on ease of implementation (e.g., manual and training is available, instrumentation for evaluation is available) and scientific merit (e.g., having multiple sites, employing a control group, tailoring the program to a population based on pre-screening), and results showed great variability across the 12 evaluated interventions. Interestingly, the grading rubric for the scientific merit of these interventions omitted the application and use of theory. As we already articulated, many of the published interventions are derived—in part—based on theoretical underpinnings that vary greatly in their focus. Perhaps the reason for this omission is that most of the interventions do indeed utilize theory; however, meta-analytic work focused on parent-related interventions to reduce cyberbullying in youth found that the effect sizes are stronger (more effective interventions) when theory was used compared to effect sizes from studies that do not utilize theory [12]. 

Myriad theories have been applied to intervention lessons; however, very few utilize tenets specific to learning theory. We want to explicate that we are not implying intervention participants are not learning valuable skills that have ramifications for behavior, nor are we implying that intervention lessons are not using pedagogy consistent with learning. However, our perspective is that most cyberbullying interventions that use theory do not adequately utilize all learning theory postulates. Using the GLM as a guide (Figure 1), we will discuss interventions that have utilized a portion of learning theory in their lessons—to great success. 

GLM delineates that after repeated learning and practice, one class of learned constructs consists of perceptual and cognitive constructs, which include perceptual and expectation schemata, beliefs, and scripts. We argued that there is a rich literature in the cyberbullying domain that maps onto these learning constructs, such as anonymity, online disinhibition, hostile attribution biases, and normative aggressive beliefs. For instance, Media Heroes [13] and ConRED [18] include lessons that outline the legal and social consequences to perpetrating online harm—consequences that many youths may not be aware of. If a student in these, and other, programs learn of the consequences of cyberbullying others, then they may learn to expect those ramifications, which fits into GLM theorizing. Focused on anonymity and online disinhibition, emerging adult participants in the You’re Not Anonymous cyberbullying intervention group received training on how their online behaviors are not as anonymous as one would believe and had a significant decrease in self-reported anonymity perceptions compared to participants in the control condition, and anonymity perceptions predicted cyberbullying perpetration several months later [102]. Finally, although we are unaware of cyberbullying interventions that specifically target normative aggressive beliefs or hostile attribution biases directly, interventions that target aggression (more broadly) by helping youth identify and tackle aggression in their classroom and fostering a safe school environment through social skills training and group work have been shown to effectively curb cyberbullying perpetration for youth in the intervention program compared to youth not in the program (ViSC [103]). 

Next, the cognitive-emotional constructs outlined in GLM include attitudes and stereotypes, which map onto cyberbullying attitudes and stereotypes. The video intervention to reduce cyberbullying created by Doane and colleagues [15] has implications for these GLM tenets. Emerging adults were presented with intervention materials that included (a) news stories of youth who committed suicide after being cyber-victimized, (b) information about what cyberbullying is and various risk and outcome factors to increase cyberbullying knowledge, and (c) acted vignettes of common cyberbullying events. Results showed that cyberbullying attitudes significantly decreased for participants in the intervention group compared to participants in the control group. 

Finally, GLM posits that affective habituation, conditioned emotions, and affective traits are emotional constructs learned after repeated learning. We then argued that empathy, desensitization, and various other personality and motivational variables (e.g., revenge motivation, anger, neuroticism) can be organized within this class of variables. Many interventions (e.g., Media Heroes [13]; the KiVA program [104]) utilize empathy training as one, of many, key component to reduce cyberbullying. 

Overall, several meta-analyses show the efficacy of cyberbullying interventions at reducing participant cyberbullying behavior, and the specific lessons across these interventions target processes delineated by GLM [22] as applicable to cyberbullying perpetration. However, much more work is needed. Here we offer some important recommendations that necessitate future work:

First, the BGCM and GLM both emphasize positively reinforced learning as an important mechanism in cyberbullying development. Therefore, it is prudent that interventions incorporate the entities that can reinforce or punish cyberbullying actions. Indeed, Barlett (2019) noted that cyberbullying prevention necessitates a multi-entity approach—parents, peers, the school, and the individual are all necessary to stop cyberbullying perpetration. Fortunately, many intervention programs include parents [18] and peers [105] in their approach. However, most of the lessons given to the larger community focus on cyberbullying knowledge (definitions, effects, motivations, etc.) without focusing on the reinforcement. We argue that interventions that target many interested groups are wonderful and cyberbullying knowledge is important. Perhaps additional information regarding how to appropriate reinforce or punish cyberbullying actions can help make these interventions more effective. However, this is an empirical question that requires future work. 

Second, the evidence we presented to support the contention that cyberbullying is a learned social behavior suggests that cyberbullying interventions need to be administered to participants who are in a developmentally appropriate age range—old enough to understand the content of the intervention but young enough to begin to have access to online technologies and start using social media independent of their parents/guardians. For instance, Englander [106] showed that youth aged 8–11 started cyberbullying others, especially when they owned a cellular phone. Much thought will be needed to address the issue of participant age if such a cyberbullying curriculum can be tailored to elementary-school-aged youth. 

Third, theories used to predict cyberbullying argue for myriad potential mediating variables that describe why cyberbullying perpetration occurs. Cyberbullying perpetration interventions target several of these, such as attitudes [15], empathy [13], aggressive behaviors [103], and others. However, several other potential mediators are left understudied. For example, we are unaware of any interventions that target emotional intelligence, BIMOB, stereotypes, or revenge or fun-seeking motives—all explicated by GLM (Figure 1). Perhaps several of these variables change as a function of intervention lessons. For example, showing participants stories about youth who committed suicide after cyber-victimization incidents [15] may decrease fun-seeking motivations for cyberbullying others; however, this—and other variables—have not been tested. We understand that intervention specialists cannot measure every possible mediator, and we are not advocating for any one intervention or study to do that. Our position, though, is that other mediating variables that predict cyberbullying perpetration and can be the target of intervention lessons need to be studied in future work. 

## 9. Future Work

While the notion of using learning theory and research to understand and ultimately reduce cyberbullying perpetration is not novel, there is surprisingly a few studies that have tested and validated such claims. There is a much room for future work, and we will elucidate some of these ideas here. First, we are unaware of any studies that have tested hypotheses important for exemplifying cyberbullying learning, such as: (a) the number of proximate GLM trials needed to develop cyberbullying knowledge structures in the distal GLM, (b) whether reinforcement/punishment moderates these learning tenets, and (c) the temporal ordering of variables within GLM to cause cyberbullying behavior to be learned. To answer these important questions, researchers would need to conduct a longitudinal study that samples youth participants who have never cyberbullied another nor been cyber-victimized at Wave 1 and then assess their cyberbullying-related knowledge structures and behavior in subsequent waves. Such a study design would identify youth who have (versus have not) engaged in cyberbullying behaviors and examine the longitudinal predictors of such learning. Such a study is procedurally questionable; however, if such a study can be conducted then many questions can be answered.

Second, we believe that the importance of reinforcement in the BGCM is an understated and understudied postulate that is desperately in need of future work. A paucity of research has examined the peer/family reinforcement positive correlation with cyberbullying [25], and much more longitudinal work is needed. Conversely, operant conditioning (and GLM/BGCM) posits that punishment may be helpful to reduce the likelihood of subsequent cyberbullying; however, we few studies have tested these predictions. We perceive punishment from parents/guardians as one method; however, punishment from peers, the victim’s retaliation, and oneself are all possible. For instance, the Media Heroes cyberbullying curriculum includes lessons specific to teaching youth about the legal consequences of cyberbullying [13]; however, while Media Heroes is effective at reducing cyberbullying, we are unaware of any research specifically delving into that component of the curriculum. Future work is needed to test whether informing participants about the legal ramifications is deterrent enough to reduce cyberbullying via operant conditioning tenets. 

Finally, a state-based paradigm for measuring cyberbullying (or cyber-aggression) is desperately needed. To date, scientists have relied on vignettes to experimentally manipulate some aspect of a hypothetical cyberbullying event, which should not be interpreted as behavior. The challenge for scholars is to attempt to quantify online statements as hostile when those same statements are difficult to interpret given the lack of tone, sarcasm, emphasis, etc., that is common across online communication [107]. Research tools are getting closer to accomplishing this goal. For instance, Rezvani and Beheshti [108] validated computer software to detect cyberbullying, and future work should attempt to utilize these innovative programs to measure cyberbullying behavior in the moment. 

## 10. Overall Conclusions

Scholars, school administrators, teachers, doctors, parents, and school pupils all recognize that cyberbullying perpetration and victimization are important topics. The purpose of the current manuscript was to argue that cyberbullying can be explained using learning theory—the general learning model and Barlett Gentile cyberbullying model. However, we also aimed to highlight the need for continued research that applies these, and other, theories to cyberbullying. Overall, we hope that a better understanding of the processes germane to cyberbullying can help guide intervention efforts that successfully reduce cyberbullying perpetration. 

## Figures and Tables

**Figure 1 children-10-00325-f001:**
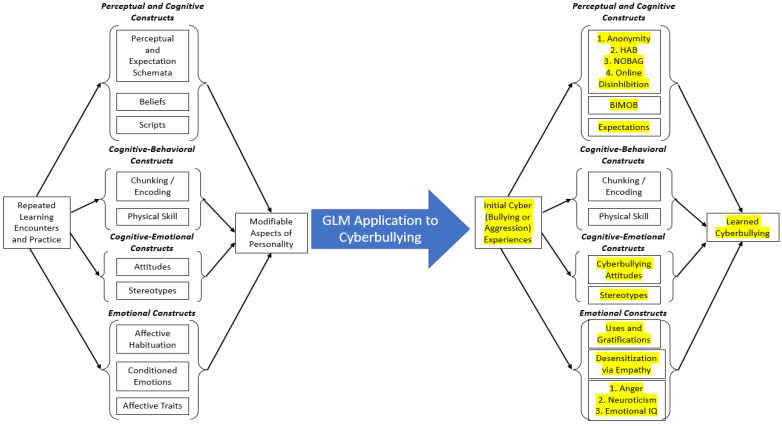
GLM applications to cyberbullying perpetration. Adapted with permission from Gentile et al. (2009). Copyright 2009 Sage Publications.

**Figure 2 children-10-00325-f002:**
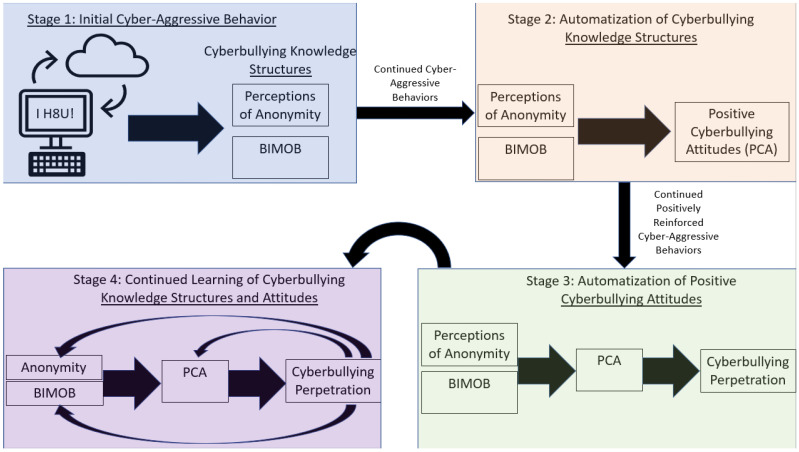
Barlett Gentile Cyberbullying Model.

**Figure 3 children-10-00325-f003:**
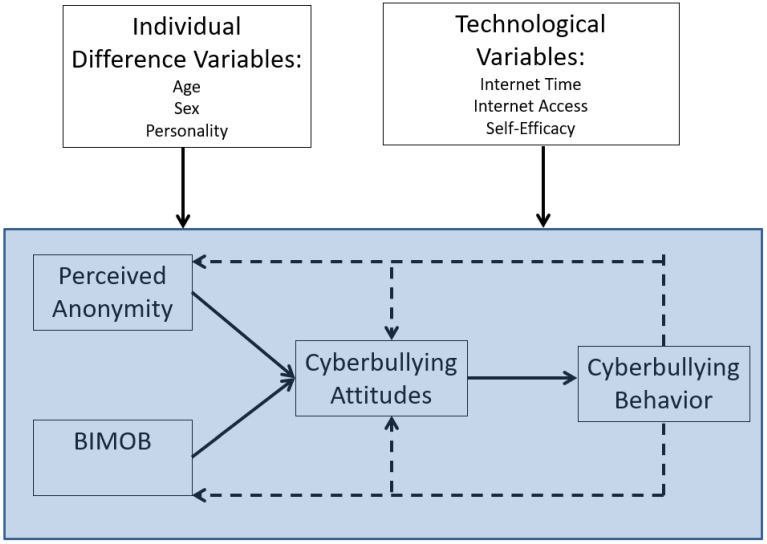
Conceptual Moderating Influence on BGCM Processes.

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
