# Peer review of "Cyberbullying as a Learned Behavior: Theoretical and Applied Implications"

_children, 2023, doi:10.3390/children10020325_

Round 1
Reviewer 1 Report
Dear Author,
my main concern is the lack of a proper description of the method of analysis in the text. We need to find out how the texts for analysis were selected and how they were analyzed.
Secondly, the main text should be shorter and more specific, and the conclusions should be longer and indicate how to translate the analysis results into practice: preventive programs, for example.
Minor notes: cyberbullying is not anti-social behaviour; please don't call it that (e.g. abstract). Please consider whether the subject of the text is cyberbullying perpetration or victimisation. Or both? Please be consistent.
Kind regards
Author Response
1. My main concern is the lack of a proper description of the method of analysis in the text. We need to find out how the texts for analysis were selected and how they were analyzed.
This paper is not a review article in the sense that a systematic review was undertaken. True, had a systematic review been conducted here then I would have to go through the procedure of properly documenting article retrieval processes, how my conclusions were drawn, etc. Instead, this is a theoretical paper and, therefore, did not have to go through those rigors.
2. Secondly, the main text should be shorter and more specific, and the conclusions should be longer and indicate how to translate the analysis results into practice: preventive programs, for example.
We retained the introduction (main text) of the paper. Indeed, the purpose of the paper is to argue for learning theories in cyberbullying and we did not want to remove important information. Moreover, representatives from the journal continued to ask for more content, which we included in the main text. We did, however, expand the intervention section of the latter part of the paper drastically. This should help articulate better how we can take theory and translate it to prevention efforts.
3. Minor notes: cyberbullying is not anti-social behaviour; please don't call it that (e.g. abstract). Please consider whether the subject of the text is cyberbullying perpetration or victimisation. Or both? Please be consistent.
The paper focuses on cyberbullying perpetration, not victimization. I rewrote portion of the paper to reflect that and made sure to cite studies that focused on perpetration only. Although we disagree with this reviewer and view cyberbullying perpetration as an antisocial behavior, it is not our position here or in our paper to articulate that. Thus, we removed the word “antisocial” from the abstract and one other place in the paper.
Reviewer 2 Report
1. This article is not a review. It is a theoretical article. No hypotheses are tested, but a number of interesting hypothesis re the learning of cyberbullying are offered, with backing literature, and these are eventually integrated into one particular model. This reads very much like a chapter in a book.
2. I marked "no" for inappropriate self-citation, but it should be noted that this article cites work by one particular group of researchers very frequently. It also cites other work, so there is a balance.
3. The language of this article is ponderous. The article does not read smoothly. Every time I encountered the phrase "cyberbulling perpetration", it threw me off. Why not just "cyberbullying". We are unlikely to be speaking of things that do not happen (are not perpetrated). However, I will honor the author's right to chose the level of his language.
4. There are small language errors here and there - e.g., line 8 (curricula is a plural noun) and line 31 (evidenced is appropriate), as well as lines 21-22. These errors are easily corrected.
5. Table 1. This table needs to be clarified with better footnotes. Were any of the effect sizes of the eta squared family? or were they of the Cohen's d family? (OR is clearly noted). What is their relevance to this article?
As well, this table is not particularly useful as it does not describe the types of interventions being evaluated. As well, the author does not mention overall conclusions reached in the meta-analyses in any detail.
6. Lines 178-182: the author should be discussing correlation not causation. So the opening statement of that paragraph is not appropriate (while the second sentence is OK).
7. Lines 204-205: the author seems to be offering only one viewpoint on an issue that has been argued from both sides (influence of media on cyberbullying).
8. The author is obviously very familiar with GLM, BCGM, and BIMOB. These acronyms are not very useful to the reader who is not acquainted with the concepts. Especially in Figure 2, BIMOB needs to be explained.
9. What is a "Theoretical and Applied Implications Perspective" (in the title)?
Overall Comment
This article could be improved by a crisper writing style and a clearer expression of its purpose.
Author Response
- This article is not a review. It is a theoretical article. No hypotheses are tested, but a number of interesting hypothesis re the learning of cyberbullying are offered, with backing literature, and these are eventually integrated into one particular model. This reads very much like a chapter in a book.
This is true. In fact, we noted in the paper that we will not be presenting a systematic review of the literature. The value in our paper, as we see it, is in the theoretical derivation of cyberbullying as the reviewer noted. We hope that this paper will serve as a resource for scholars to think more deeply about learning theory and cyberbullying, interventionalists who are looking for ways to improve or create cyberbullying prevention techniques and lessons, and the general public who is interested in learning why people engage in cyberbullying behavior. To help clarify this we removed the word “review” from the paper. One final note: although this paper is not a review of the literature (in terms of using systematic review procedures), the paper does fit within the scope of the journal and the content of the special issue.
- I marked "no" for inappropriate self-citation, but it should be noted that this article cites work by one particular group of researchers very frequently. It also cites other work, so there is a balance.
We removed several of the citations from the Barlett group. Now, we mostly cite the Barlett work predominantly in the BGCM section, as appropriate. We replaced several Barlett citations with other scholar’s work throughout the paper.
- The language of this article is ponderous. The article does not read smoothly. Every time I encountered the phrase "cyberbulling perpetration", it threw me off. Why not just "cyberbullying". We are unlikely to be speaking of things that do not happen (are not perpetrated). However, I will honor the author's right to chose the level of his language.
Thank you for this comment. Based on another reviewer’s comments and work in this area, we have to explicate “cyberbullying perpetration” to distinguish it from “cyber-victimization.”
- There are small language errors here and there - e.g., line 8 (curricula is a plural noun) and line 31 (evidenced is appropriate), as well as lines 21-22. These errors are easily corrected.
We reread and thoroughly edited the paper to (hopefully) avoid this issue.
- Table 1. This table needs to be clarified with better footnotes. Were any of the effect sizes of the eta squared family? or were they of the Cohen's d family? (OR is clearly noted). What is their relevance to this article? As well, this table is not particularly useful as it does not describe the types of interventions being evaluated. As well, the author does not mention overall conclusions reached in the meta-analyses in any detail.
We removed Table 1. Upon review, we noticed that Table 1 did not add much to the paper and created more confusion that clarity. Thus, we simply cited the work that was in Table 1.
- Lines 178-182: the author should be discussing correlation not causation. So the opening statement of that paragraph is not appropriate (while the second sentence is OK).
We changed the sentence to address this issue.
- Lines 204-205: the author seems to be offering only one viewpoint on an issue that has been argued from both sides (influence of media on cyberbullying).
We clarified this in the text. We are actually offering two view-points. The first is that media violence exposure is directly related to cyberbullying and the second is that media violence is indirectly related to cyberbullying via aggression-related variables (e.g., normative aggressive beliefs).
- The author is obviously very familiar with GLM, BCGM, and BIMOB. These acronyms are not very useful to the reader who is not acquainted with the concepts. Especially in Figure 2, BIMOB needs to be explained.
BIMOB is explained in the text (section 5.1), which occurs before Figure 2. Also, we made sure to denote the meaning of these acronyms early in the paper to avoid confusion. Thus, we did not address this further.
- What is a "Theoretical and Applied Implications Perspective" (in the title)?
This title was accidentally retained during the editing by using track changes that were not accepted when it was transferred over to the journal formatting. We changed this.
- Overall Comment: This article could be improved by a crisper writing style and a clearer expression of its purpose.
Thank you for the review and comments. We feel the paper is stronger than before.
Round 2
Reviewer 1 Report
Thank you, now I accept the article.